# Evaluation of a lateral flow immunochromatographic assay for detecting cryptococcal antigens in bronchoalveolar lavage fluid from HIV-negative patients with pulmonary cryptococcosis

Xin Chen,[1,2] Zhanwei Wang,[1] Chun Di,[1] Feifei Zhang,[1] Janhua Yu,[3] Jiangfan Ran,[2] Ke Wang,[4] Zhixiang Li,[5] Huiying Zhao,[6] Hongbin Chen,[1] Qi Wang,[1] Hui Wang[1]

**ABSTRACT** Cryptococcal antigen lateral flow immunochromatographic assay (CrAg-LFA) is extensively utilized for the diagnosis of cryptococcosis in serum and cerebrospinal fluid. However, its application in bronchoalveolar lavage fluid (BALF) has been relatively underexplored. This study aimed to assess the diagnostic performance of the CrAg-LFA in BALF from human immunodeficiency virus (HIV)-negative patients with pulmonary cryptococcosis (PC) over an 11-year period in a tertiary hospital in China and to compare it with serum samples. Among 514 clinically suspected PC patients at a tertiary hospital (January 2013 to February 2024), 42 were diagnosed with PC. BALF CrAg-LFA demonstrated a sensitivity of 88.1%, a specificity of 99.6%, a positive predictive value (PPV) of 94.9%, and a negative predictive value (NPV) of 98.9%. In contrast, the serum CrAg-LFA showed a sensitivity of 90.5%, a specificity of 99.6%, a PPV of 95.0%, and an NPV of 99.2%. The combination of BALF and serum CrAg-LFA enhanced both sensitivity and NPV to 100%, while maintaining high specificity (99.1%) and PPV (91.3%). One patient tested negative for CrAg-LFA in serum but positive in BALF. These findings underscore the excellent diagnostic accuracy of both specimen types for CrAg detection in HIV-negative PC patients and offer complementary diagnostic utility in specific clinical scenarios, such as cases with false-negative results. When bronchoscopy is indicated, adjunctive BALF CrAg testing can significantly enhance PC diagnosis, especially in patients with negative serum CrAg but strong clinical suspicion of infection.

**IMPORTANCE** This study demonstrates the significant diagnostic value of the cryptococcal antigen lateral flow immunochromatographic assay (CrAg-LFA) for pulmonary cryptococcosis (PC) in human immunodeficiency virus (HIV)-negative patients. We found that CrAg-LFA performed on bronchoalveolar lavage fluid (BALF) was highly effective, with a sensitivity of 88.1% and a specificity of 99.6%. Importantly, combining BALF and serum tests increased sensitivity and negative predictive value to 100%, providing a more reliable diagnostic approach. These findings highlight the utility of BALF CrAg-LFA in facilitating early and accurate PC diagnosis, which can significantly improve patient outcomes in non-HIV populations. This research offers critical insights into optimizing diagnostic strategies for this challenging condition.

**KEYWORDS** bronchoalveolar lavage fluid, cryptococcal antigen, pulmonary cryptococcosis, lateral flow immunochromatographic assay

Pulmonary cryptococcosis (PC) is an opportunistic invasive fungal disease that is widely prevalent in individuals infected with the human immunodeficiency virus (HIV) and those with impaired immune systems and can also cause severe consequences in immunocompetent people. The primary pathogens responsible for PC are

**Peer Reviewer** Fangyou Yu, Shanghai Pulmonary Hospital of Tongji University School of Medicine, Shanghai, China

Address correspondence to Qi Wang, wangqi_pkupph@163.com, or Hui Wang, whuibj@163.com.

Xin Chen, Zhanwei Wang, and Chun Di contributed equally to this article. Author order was determined by drawing straws.

The authors declare no conflict of interest.

*Cryptococcus neoformans* and *Cryptococcus gattii*, which are prevalent in the environment, particularly in soil and other environments. *Cryptococcus* can enter the human body through inhalation and initially remains stationary in the lungs. When the immune system is weakened, pulmonary infections and distant infiltration of the central nervous system (CNS) subsequently occur (1).

Early diagnosis of PC is challenging due to the atypical clinical symptoms and imaging features, relying heavily on histopathological and culture-based methods. However, these methods have limited sensitivity, and pathogenic *Cryptococcus* may be overgrown and obscured by faster-growing respiratory colonizing bacteria, causing false-negative results. Consequently, the diagnosis of PC is frequently delayed, leading to severe disease progression (2). Therefore, numerous studies have focused on the use of cryptococcal antigen (CrAg) detection as an early diagnostic tool for PC in recent years.

Compared with histopathology and culture-based methods, latex agglutination and enzyme immunoassay, which have been widely used for CrAg detection in serum and cerebrospinal fluid (CSF) of patients with cryptococcosis, exhibited higher sensitivity and specificity. However, these methods are cumbersome, time-consuming, and not suitable for routine testing in bronchoalveolar lavage fluid (BALF) for the early diagnosis of pulmonary PC. In 2009, the lateral flow immunochromatographic assay (LFA) was introduced as a novel CrAg detection method. Supported by numerous studies, the LFA was characterized by its simplicity, rapidity, sensitivity, and high specificity. Since 2011, it has been widely adopted for CrAg testing in serum, CSF, BALF, and urine (3–6).

Recent data indicate that the prevalence of PC among HIV-negative patients has been increasing (7, 8). Currently, CrAg-LFA testing for PC diagnosis primarily relies on serum and CSF specimens (9). However, studies have shown that the sensitivity of the LFA may be relatively lower in serum samples from PC patients without diffuse lesions (2). In such cases, BALF may serve as a critical diagnostic alternative, particularly when lung tissue is unavailable. Nevertheless, research on the diagnostic value of the CrAg-LFA in BALF for HIV-negative PC patients remains limited, highlighting the need for further investigation to elucidate its potential clinical applications. This study evaluated the diagnostic efficacy of the BALF CrAg-LFA test in HIV-negative PC patients by analyzing CrAg-LFA outcomes from both BALF and serum samples collected over an 11-year period in Beijing, China.

## MATERIALS AND METHODS

### Clinical data screening

This study represents a prospective analysis of patients suspected of having PC who attended a tertiary hospital in Beijing from January 2013 to February 2024. Inclusion criteria comprised non-HIV patients who demonstrated PC imaging features (including both single or multiple nodules, segmental consolidation, cavitary lesions, a diffuse miliary pattern, or mixed patterns, especially located in the peripheral area of the lung) with no prior antifungal therapy and complete baseline demographic, clinical, and laboratory data. Exclusion criteria involved patients with uncertain diagnoses or a history of cryptococcal infection. Ultimately, 514 patients were included. Their clinical data were collected, and each patient's computed tomography (CT) scan was independently assessed by a radiologist.

The criteria for the diagnosis of PC were based on the revised 2019 consensus definitions of invasive fungal disease issued by the European Organization for Research and Treatment of Cancer and the Mycoses Study Group Education and Research Consortium (9). Immunosuppressed populations were defined according to the 2020 CHEST consensus statement (10).

### Pathological examination

Histopathological or cytological specimens were collected from patients (including transbronchial needle aspiration, transbronchial lung biopsies, and CT-guided

percutaneous lung biopsies). The samples were embedded, sectioned, and analyzed by specialized pathology technicians blinded to their clinical origin. All specimens underwent H&E staining, periodic acid-Schiff (PAS) staining, and special stains (e.g., Grocott's methenamine silver and mucicarmine). *Cryptococcus* positive cases were characterized by round or ovoid yeast-like fungi (4–10 µm in diameter) with narrow-based budding ("teardrop" morphology) and a surrounding hollow halo (representing the mucopolysaccharide capsule).

## Sample collection and testing

BALF was collected from patients after chest CT-guided localization and local anesthesia with 2% lidocaine. A flexible bronchoscope (Olympus BF-260 series, Japan) was positioned in the target segment. A total of 100 mL of room-temperature saline was instilled in 3–5 aliquots (20–50 mL each), followed by suction at <100 mmHg after each aliquot. The minimum recovered volume was ≥5% of the instilled volume, with optimal recovery >30% (11).

A 3 mL blood sample was drawn before antifungal treatment into a separator gel tube with negative pressure, and the serum was separated by centrifugation at $3,000 \times g$ for 10 min.

CSF was obtained via lumbar puncture between the C3 and C4 vertebrae under local anesthesia, provided there were no contraindications. The CSF was sent to the microbiology laboratory within 30 min or directly injected into blood culture bottles with neutralized antibiotics (BioMerieux, France).

BALF and CSF samples were cultured to detect the presence of *Cryptococcus* and were also subjected to Gram staining and India ink staining for the identification of round, encapsulated yeast. In line with the instructions provided by the CrAg-LFA kit (Immuno Mycologics [IMMY], USA), CrAg was tested by placing the test strips into 40 µL of serum, BALF, or CSF. After 10 min, the results were interpreted, with a red line appearing in both the control and detection zones, indicating a positive outcome. For serum specimens testing CrAg-positive were then serially diluted to determine CrAg titers, with a titer ≤1:5 defined as weakly positive. However, CrAg titer testing was not performed in BALF or CSF specimens due to procedural variations in sample collection that would compromise titer reliability. (A weakly positive BALF CrAg-LFA result was defined by a test line intensity lower than that of the control line.)

## Metagenomic next-generation sequencing (mNGS) analysis

We also performed mNGS tests on some BALF, blood, or biopsy samples for *Cryptococcus*. Following sample processing, DNA was extracted using the DP316 kit (Tiangen Biotech, China), and RNA was isolated using a commercial kit (Microgene Biotech, China). The libraries were built using the QIAseq Ultralow Input Library kit for Illumina (Qiagen, Hilden, Germany), with quality assessed using a Qubit (Thermo Fisher Scientific, MA, USA) and an Agilent 2100 Bioanalyzer (Agilent Technologies, Palo Alto, USA). Sequencing was performed on the NextSeq 550 platform (Illumina, San Diego, USA). For bioinformatics analysis, raw data were processed by removing adapter sequences, short (<36 bp), low-quality ($Q < 30$), and low-complexity reads using bowtie2. Human sequences were filtered by mapping to the hg38. The remaining reads were finally aligned to the NCBI Microbial Genome Databases (https://www.ncbi.nlm.nih.gov) using Burrows-Wheeler Aligner, with species annotation determined by least common ancestor analysis. A result was considered positive if at least three non-overlapping reads were mapped to the species level and absent in the no-template control (NTC) or the detected reads were ≥10-fold more than those in the NTC.

## Statistics

Statistical analysis was performed using SPSS 25 (IBM Corp., Armonk, NY, USA). Continuous variables with a normal distribution were expressed as mean ± standard deviation

(SD), and group comparisons were conducted using the independent samples *t*-test. For non-normally distributed continuous variables, data were expressed as median (interquartile range), and group comparisons were performed using the Mann-Whitney *U* test. Categorical variables were presented as frequencies (percentages), with intergroup comparisons performed using the chi-square test or Fisher's exact test, as appropriate. A *P* value of <0.05 was considered statistically significant.

## RESULTS

### Clinical characteristics of PC patients

In this study, 42 patients were diagnosed with PC, consisting of 19 males and 23 females, with an age range of 26–82 years (mean ± SD: 54.7 ± 13.8). The diagnosis was established through multiple methods: 8 (19.0%) patients were identified via positive culture, India ink preparation, or Gram staining of BALF (one of these patients also had positive CSF cultures and smears); pathological confirmation was obtained in 10 (90.9%) patients (11 patients in total underwent pathological evaluation); and 8 (47.1%) patients were confirmed to be *Cryptococcus* positive through mNGS testing of BALF, blood, or biopsy samples (17 patients underwent mNGS testing). The remaining cases were diagnosed based on a combination of clinical manifestations consistent with PC, characteristic imaging findings, positive serum CrAg testing, and demonstrated treatment response.

Twenty-five (59.5%) patients were symptomatic. The most common symptoms included fever, cough, dyspnea, and headache; altered mental status and hematuria were occasionally observed. A significant proportion of patients, 34 (80.9%), had one or more underlying conditions. Laboratory analyses indicated that most patients exhibited normal leukocyte and C-reactive protein (CRP) levels, whereas lymphocyte counts in BALF were elevated in 90% of cases.

All the PC patients underwent chest CT, which revealed diverse lesion characteristics. Specifically, nodules alone were observed in 8 (19.0%) patients, diffuse inflammatory shadows alone in 15 (35.7%), and a combination of both in 19 (45.2%). Additionally, lesions were localized to unilateral lobes in 21 (50.0%; left/right = 4/17) patients, while the remaining 21 (50.0%) showed bilateral involvement.

Among PC patients, 17 (40.5%) patients were immunosuppressed: 16 had documented autoimmune diseases with chronic immunosuppressive therapy, and 1 patient was undergoing chemotherapy for hematological malignancy. A history of avian exposure (defined as contact with pigeons through domestic keeping or frequenting congregation areas) was reported by 15 (35.7%) patients. Notably, a significant inverse association was observed between these two factors: the majority of immunosuppressed patients (16/17; 94%) had no avian exposure history, whereas most patients with avian exposure (14/15; 93%) were immunocompetent (*P* = 0.001). This finding suggests that avian exposure may be a potential etiological factor for PC in immunocompetent hosts. These demographic and clinical data are summarized in Table 1.

### Comparison of the sensitivity and specificity of the CrAg-LFA in BALF and serum samples

Table 2 summarizes the sensitivity and specificity of the CrAg-LFA performed on BALF and serum. The BALF CrAg-LFA demonstrated a sensitivity of 88.1% and a specificity of 99.6%, while the serum CrAg-LFA exhibited a sensitivity of 90.5% and a specificity of 99.6%. Although the sensitivity of serum CrAg-LFA was slightly higher than that of the BALF CrAg-LFA, the difference was not statistically significant (*P* = 0.267). Notably, the specificity for both specimen types was equally high at 99.6%, indicating that both BALF CrAg-LFA and serum CrAg-LFA tests are equally reliable for diagnosing PC. Additionally, both specimen types also showed high positive predictive value (PPV) and negative predictive value (NPV): BALF CrAg-LFA had a PPV of 94.9% and an NPV of 98.9%, while serum CrAg-LFA had a PPV of 95.0% and an NPV of 99.2%. The NPV was particularly noteworthy in both tests.

**TABLE 1** Clinical characteristics of 42 PC patients

| Patient variable | Data |
| --- | --- |
| Age, y | 54.7 ± 13.8 ($\overline{x} \pm s$) |
| Sex, male:female | $n$ = 19:23 |
| Laboratory findings | |
| Positive pathology | $n$ = 10/11 (90.9%) |
| Positive BALF/blood/biopsy mNGS | $n$ = 8/17 (47.1%) |
| Positive BALF cultures/smears | $n$ = 8/42 (19.0%) |
| Positive BALF and CSF cultures | $n$ = 1/34 (2.9%) |
| Leukocytosis (>9.5 × $10^9$/L) | $n$ = 10/42 (23.8%) |
| Elevated CRP (>10 ng/L) | $n$ = 12/42 (28.6%) |
| Elevated BALF lymphocytes (>10%) | $n$ = 36/40 (90.0%) |
| Symptoms | |
| Symptomatic[a] | $n$ = 25 (59.5%) |
| Asymptomatic[b] | $n$ = 17 (40.5%) |
| CT images | |
| Two lungs involved | $n$ = 21 (50.0%) |
| Unilateral lung involved (left/right) | $n$ = 21 (50.0%) (4/17) |
| Nodules | $n$ = 8 (19.0%) |
| Diffuse inflammatory shadows | $n$ = 15 (35.7%) |
| Diffuse inflammatory shadows with nodules | $n$ = 19 (45.2%) |
| Underlying diseases[c] | |
| Respiratory diseases | $n$ = 12 (28.6%) |
| Digestive system diseases | $n$ = 10 (23.8%) |
| Cardiovascular system diseases | $n$ = 21 (50.0%) |
| Endocrine system diseases | $n$ = 21 (50.0%) |
| Autoimmune diseases | $n$ = 18 (42.9%) |
| Renal diseases | $n$ = 2 (4.8%) |
| Hematological diseases | $n$ = 4 (9.5%) |
| Cancers | $n$ = 4 (9.5%) |
| Avian exposure history[d] | $n$ = 15 (35.7%) |
| Immunosuppressed cases | $n$ = 17 (40.5%) |

[a]These patients presented with symptoms such as fever, cough, dyspnea, headache, altered mental status, and hematuria during the medical visit.
[b]These patients have imaging features of PC without associated clinical symptoms.
[c]Some people have more than one underlying condition.
[d]Avian exposure history was defined as contact with pigeons, such as through domestic keeping or frequenting areas where they congregate.

Defining a positive combined test result as positivity in either BALF or serum (or both), and a negative result as negativity in both, the combination of BALF with serum CrAg-LFA increased both sensitivity and NPV to 100%, while also demonstrating high specificity (99.1%) and PPV (91.3%).

## Relationships between CT characteristics, CrAg intensity, and clinical symptoms in 42 PC patients

Statistical analysis revealed significant correlations between imaging findings and CrAg titers. Patients with unilateral lung lesions (15/21, 71.4%) were significantly more likely to have low serum CrAg titers (≤1:20) compared to those with bilateral involvement ($P$ = 0.005) (Fig. 1A). Moreover, 4/5 (80%) false-negative and 5/5 (100%) weakly positive BALF CrAg results were associated with serum CrAg titers ≤ 1:20 (Fig. 1B). These low-titer BALF results (false-negative: 4/5, 80%; weakly positive: 4/5, 80%) were observed in PC patients with unilateral lung lesions (Fig. 1C).

In terms of clinical presentation, 13/15 (87%) PC patients with diffuse inflammatory shadows were symptomatic, while 6/8 (75%) patients with solely nodular lesions were asymptomatic (cases with imaging features of PC without associated clinical symptoms)

**TABLE 2** Sensitivity and specificity of the CrAg-LFA in this study[d]

| | Confirmed PC diagnosis (n) | | Sensitivity (%) | Specificity (%) | PPV[a] (%) | NPV[b] (%) |
|---|---|---|---|---|---|---|
| | Positive | Negative | | | | |
| BALF CrAg-LFA | | | | | | |
| Positive | 37 | 2 | 88.1 | 99.6 | 94.9 | 98.9 |
| Negative | 5 | 470 | | | | |
| Serum CrAg-LFA | | | | | | |
| Positive | 38 | 2 | 90.5 | 99.6 | 95.0 | 99.2 |
| Negative | 4 | 470 | | | | |
| BALF and serum CrAg-LFA combined test[c] | | | | | | |
| Positive | 42 | 4 | 100 | 99.1 | 91.3 | 100 |
| Negative | 0 | 468 | | | | |

[a]PPV, positive predictive value.
[b]NPV, negative predictive value.
[c]The positive result is defined as BALF+/serum+ or BALF+/serum− or BALF−/serum+ whereas a negative result is defined as BALF−/serum−.
[d]The analysis encompassed 514 cases.

($P = 0.012$). The type of lesion, however, did not correlate with the immune status of the patients ($P = 0.712$). Additionally, asymptomatic PC patients were significantly more likely to have serum CrAg titers of ≤1:40 (14/17, 82%; $P = 0.044$; Fig. 2).

## Consistent pattern of weakly positive CrAg results between BALF and serum samples

Analysis of CrAg intensity revealed a consistent pattern and a complementary diagnostic relationship between BALF and serum samples (Table 3). Among the five cases with weakly positive BALF CrAg, four had matching weakly positive serum titers (≤1:5), and one was serum-negative. Similarly, out of the 12 patients with weakly positive serum CrAg results, 10 showed weakly positive ($n = 4$) or negative ($n = 6$, including two confirmed serum false positives) BALF-CrAg results; the remaining two patients, despite exhibiting positive BALF CrAg tests, were asymptomatic, with chest CT scans showing solitary nodules indicative of early-stage disease.

## DISCUSSION

In China, PC ranks as the third most common invasive fungal disease, with an increasing incidence and a growing proportion of non-immunosuppressed patients (2, 12). It is worth noting that the proportion of PC patients with early-stage lesions or no clinical symptoms is also on the rise, particularly since the introduction of CrAg-LFA.

The CrAg-LFA, which utilizes an LFA platform, detects free cryptococcal capsular antigen in the body fluids such as blood and CSF. This method is sensitive, simple, inexpensive, and capable of identifying all serotypes of CrAg. It does not require specialized equipment or advanced technical expertise, and results can be obtained within approximately 10 min (13). These characteristics make it well-suited for surveillance in regions with limited medical resources but high cryptococcosis prevalence.

In theory, BALF, often referred to as "lung fluid tissue," provides a direct means of assessing pulmonary infection status. It may offer superior diagnostic value over serum specimens in PC patients, particularly for small or localized lesions. However, research on the diagnostic utility of BALF CrAg-LFA in HIV-negative PC patients remains limited, with only five reports from China (Fujian, Guangdong, and Zhejiang) available to date (14–18), and their conclusions have been inconsistent. Thus, this study aimed to evaluate the diagnostic significance of BALF CrAg-LFA and compare it with serum CrAg-LFA among 514 patients with suspected PC in Beijing.

This study demonstrates that BALF CrAg-LFA and serum CrAg-LFA showed sensitivities of 88.1% and 90.5%, respectively, with specificities of 99.6% for both. Interestingly, no significant diagnostic performance difference was observed between BALF and serum samples—a finding that contrasts with previous reports by Zeng et al., Zhu et

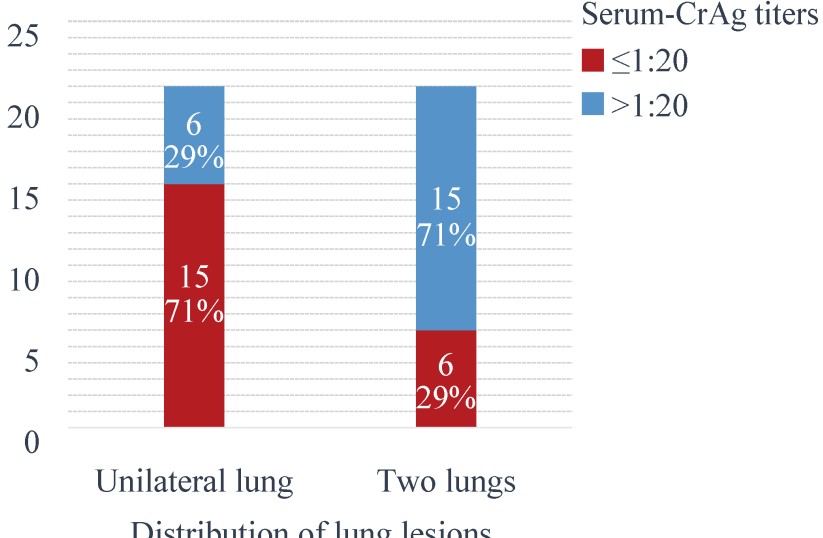

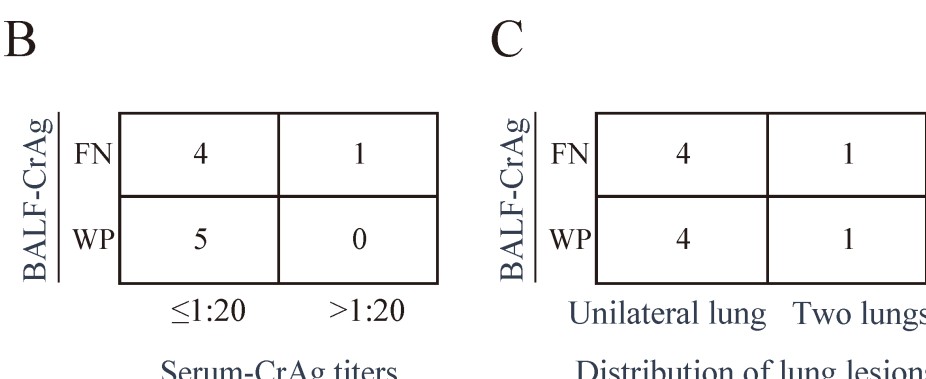

**FIG 1** Relationship between lung lesion distribution and CrAg titers in patients with PC. (A) Distribution of serum CrAg titers based on lung lesion involvement. For unilateral lung lesions, 71% of cases exhibited serum CrAg titers ≤ 1:20, while 29% showed titers > 1:20. In contrast, for bilateral lung lesions, 29% had titers ≤ 1:20 and 71% had titers > 1:20 (*P* = 0.005). (B) Outcomes of the BALF-CrAg test. Among the results, 4 out of 5 false negatives (FN) and all 5 weak positives (WP) were associated with serum CrAg titers ≤ 1:20. (C) Association between lung lesion distribution and BALF-CrAg test outcomes. Unilateral lung lesions were observed in 4 out of 5 FN and 4 out of 5 WP.

al., and Yang et al. (14, 15, 17). One possible explanation is that diffuse inflammatory shadows were observed in 33 (78.6%) of the PC patients in our case series. CrAg derived from such diffuse lesions may be more likely to enter the bloodstream compared to localized nodular lesions, facilitating easier detection in serum. Additionally, various factors (such as irrigation volume, recovery rate, sampling quality, transport delays, improper storage, freeze-thaw cycles, and container interference) in BALF collection and processing may contribute to the observed discrepancies across studies. The combination of BALF and serum CrAg-LFA achieved 100% sensitivity, 99.1% specificity, and 100% NPV. These results indicate that BALF-CrAg testing is particularly valuable for evaluating suspected PC cases with negative serum CrAg results, serving as a critical complementary diagnostic tool.

This study also differs from those conducted by Zeng et al. and Yamakawa et al. (15, 19), likely due to variations in the distribution of the study population. Our data indicate that more PC patients present with clinical symptoms, and both unilateral and bilateral

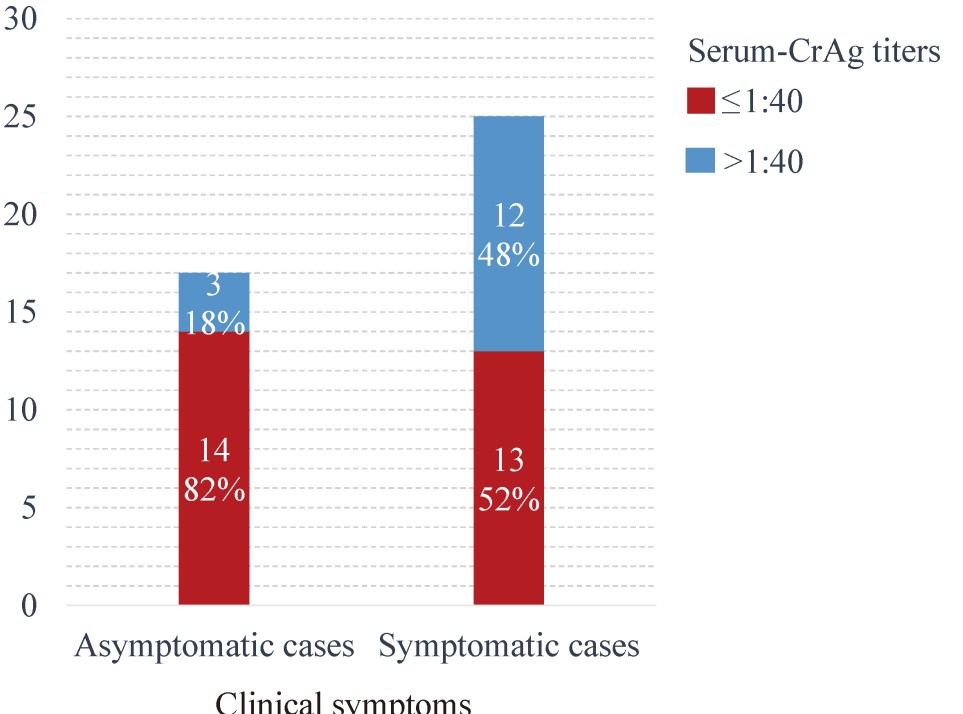

**FIG 2** Relationship between clinical symptoms and serum CrAg titers. The figure demonstrates the distribution of serum CrAg titers in asymptomatic and symptomatic patients. Asymptomatic patients were significantly more likely to exhibit low serum CrAg titers (≤1:40) than symptomatic patients (82% vs 52%; $P$ = 0.044).

lesions occur at comparable rates (Table 1). These differences may be attributable to the higher prevalence of multiple underlying conditions among these patients.

Statistical analysis of the data revealed four false positive cases in non-PC patients, two from BALF-CrAg (attributed to rheumatoid lung nodules or BALF-GM [galactomannan testing]-positive lung inflammation) and two from serum CrAg (both with lung cancer). Some researchers suggest that CrAg assays may cross-react with GM-positive microorganisms and other entities with polysaccharide antigens. Other potential interferents include detergent contamination, inaccurate dilution ratios, rheumatoid factors, and malignant tumors (15, 20, 21). Among the 42 PC patients, nine FN were identified, five for BALF-CrAg (four cases with weakly positive serum-CrAg results), and four for serum-CrAg (three cases with positive BALF-CrAg results were asymptomatic at presentation, with chest CT scans primarily demonstrating solitary nodules; the remaining one involved an immunocompromised individual who presented with a 15-day history of fever and had a weakly positive BALF-CrAg result, with chest CT showing localized patchy consolidation in the right lower lobe). FN may occur during the early stages of infection or be related to factors such as insufficient antigen dilution, low antigen titers, sample cryopreservation, or the presence of small or non-capsular cryptococci (22–25).

Based on prior experience, HIV-negative PC patients routinely undergo lumbar puncture to identify asymptomatic or subclinical CNS involvement, which may necessitate more intensive therapy (26–28). In this study, lumbar puncture with CSF analysis was performed in PC patients who met any of the following criteria: (i) neurological symptoms (fever, headache, nausea/vomiting, or altered mental status); (ii) positive serum CrAg-LFA (27, 29). Ultimately, 39/42 PC patients met these criteria. Due to various limitations (such as technical challenges, bleeding disorders, and patient refusal), CSF analysis (including culture, India ink staining, and CrAg-LFA testing) was completed in only 34 PC patients, with only one patient (who presented with headache and altered

**TABLE 3** Consistent pattern of the weakly positive CrAg results between serum and BALF samples[a]

| Patients | Serum CrAg titers | BALF CrAg intensity | Notes |
| --- | --- | --- | --- |
| Patient 1 | 1:5 | WP | |
| Patient 2 | 1:5 | P | Asymptomatic/nodules |
| Patient 3 | 1:5 | WP | |
| Patient 4 | 1:5 | P | Asymptomatic/nodules |
| Patient 5 | 1:5 | WP | |
| Patient 6 | <1:5 | N | |
| Patient 7 | <1:5 | N | |
| Patient 8 | <1:5 | N | False positive |
| Patient 9 | <1:5 | N | False positive |
| Patient 10 | <1:5 | N | |
| Patient 11 | <1:5 | N | |
| Patient 12 | <1:5 | WP | |
| Patient 13 | N | WP | |

[a]P, positive; WP, weak-positive; N, negative.

mental status) showing positive results in both culture and CrAg-LFA assays. The low detection rate of *Cryptococcus* in the CSF suggests that most patients had early-stage disease without CNS dissemination.

This study has several limitations. Although 514 suspected patients were enrolled, only 42 were definitively diagnosed with PC, resulting in a relatively small sample size for analysis. Additionally, the single-center design may introduce selection bias. Future studies should adopt a multi-center design and enroll a larger number of PC cases to validate our findings.

## Conclusions

This study is a large-scale evaluation of CrAg-LFA in BALF from HIV-negative PC patients in China. BALF CrAg-LFA showed comparable efficacy to serum CrAg-LFA. Combining BALF and serum CrAg-LFA achieved optimal performance. Given BALF's accuracy in reflecting lung infection, especially in localized lesions, we recommend BALF CrAg testing during bronchoscopy, particularly when serological tests are negative.

## ACKNOWLEDGMENTS

This study was partly supported by the National Natural Science Foundation of China (Grant No. 82172310) and the National Key Reseach and Development Program of China (Grant No. 2023YFC2307100) and Noncommunicable Chronic Diseases-National Science and Technology Major Project (No.2024ZD0532800).

## AUTHOR AFFILIATIONS

[1]Department of Clinical Laboratory, Peking University People's Hospital, Beijing, China
[2]Department of Laboratory Medicine, Foshan Fosun Chan Cheng Hospital, Foshan, China
[3]Department of Radiology, The Second Affiliated Hospital of Nanchang University, Nanchang, China
[4]Department of Quality Control, Foshan Fosun Chan Cheng Hospital, Foshan, China
[5]Department of Laboratory Medicine, The Second Affiliated Hospital, School of Medicine, The Chinese University of Hong Kong, Shenzhen, Guangdong, P. R. China Longgang District People's Hospital of Shenzhen, Shenzhen, China
[6]Department of Laboratory Medicine, Peking University Cancer Hospital Inner Mongolia Hospital, Hohhot, Inner Mongolia, China

## AUTHOR ORCIDs

Xin Chen  http://orcid.org/0009-0009-4342-1239
Qi Wang  http://orcid.org/0009-0005-0269-8435
Hui Wang  http://orcid.org/0000-0002-9824-3769

## AUTHOR CONTRIBUTIONS

Xin Chen, Data curation, Formal analysis, Investigation | Zhanwei Wang, Methodology, Resources | Chun Di, Data curation, Investigation, Software | Feifei Zhang, Resources, Validation, Visualization | Janhua Yu, Methodology, Resources, Software | Jiangfan Ran, Investigation, Software | Ke Wang, Resources | Zhixiang Li, Resources, Validation | Huiying Zhao, Investigation, Resources | Hongbin Chen, Validation, Visualization | Qi Wang, Conceptualization, Data curation, Writing – review and editing | Hui Wang, Validation

## ETHICS APPROVAL

The study was approved by the Ethics Committee of Peking University People's Hospital (approval no. 2023PHB078-002).

## ADDITIONAL FILES

The following material is available online.

Open Peer Review

**PEER REVIEW HISTORY (review-history.pdf).** An accounting of the reviewer comments and feedback.

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
