## [Reviewer comments · Microbiology Spectrum]

Microbiology Spectrum

Evaluation of a lateral flow immunochromatographic assay for detecting cryptococcal antigens in bronchoalveolar lavage fluid from HIV-negative patients with pulmonary cryptococcosis

Xi Chen, Zhanwei Wang, Chun Di, Feifei Zhang, Janhua Yu, Jiangfan Ran, Ke Wang, Zhixiang Li, Huiying Zhao, Hongbin Chen, Qi Wang, and Hui Wang

Corresponding Author(s): Hui Wang, Peking University People's Hospital

Review Timeline:

Submission Date:	July 9, 2025
Editorial Decision:	September 10, 2025
Revision Received:	October 16, 2025
Editorial Decision:	October 22, 2025
Revision Received:	November 11, 2025
Accepted:	November 17, 2025

Editor: Dhammika Navarathna

Reviewer(s): Disclosure of reviewer identity is with reference to reviewer comments included in decision letter(s). The following individuals involved in review of your submission have agreed to reveal their identity: Fangyou Yu (Reviewer #3)

Transaction Report:

DOI: <https://doi.org/10.1128/spectrum.01083-25>

Re: Spectrum01083-25 (Evaluation of a lateral flow immunochromatographic assay for detecting cryptococcal antigens in bronchoalveolar lavage fluid from HIV-negative patients with pulmonary cryptococcosis)

Dear Dr. Qi Wang:

Thank you for the privilege of reviewing your work. Below you will find my comments, instructions from the Spectrum editorial office, and the reviewer comments.

Revision Guidelines

Sincerely,
Dhammika Navarathna
Editor
Microbiology Spectrum

Reviewer #1 (Comments for the Author):

The authors present a study of the performance of a lateral flow cryptococcal antigen test using BAL fluid in non-HIV patients. The authors have either incorporated or addressed the prior reviewers' comments. I have only minor suggestions (line numbers refer to the Marked Up version):

Line 92: Incomplete sentence. Please rephrase for grammar.

Line 304: Remove the word "podocarp" as it is not widely used or understood.

Lines 231-233: The Table 2 definition of a combined positive is clear. Please add the same description to the manuscript.

Table 2: Please explain what "confirmed" means either in the legend of Table 2 or in the manuscript.

Table 3: Correct spelling of "weak" in the legend.

Reviewer #2 (Comments for the Author):

Summary of Key Findings:

This study sought to assess the performance and diagnostic value of the cryptococcal antigen lateral flow immunochromatographic assay (CrAg-LFA) (IMMY, USA) on bronchoalveolar lavage fluid (BALF) from patients with pulmonary cryptococcosis (PC) in comparison to serum. This prospective analysis included patients suspected of having pulmonary cryptococcosis (PC) receiving treatment at a tertiary Beijing hospital from Jan. 2013-Feb. 2024. 514 non-HIV patients with characteristic PC imaging and no prior antifungal therapy were included; 42 of these patients were ultimately diagnosed with PC through various means with criteria described herein. BALF CrAg-LFA had 88.1% sensitivity, 99.6% specificity, 94.9% PPV, and 98.9% NPV-values similar to the parallel serum CrAg-LFA. The authors state that combining BALF CrAg-LFA and serum CrAg-LFA results (positive result define as BALF+ and/or Serum+) increases sensitivity and NPV to 100% and suggest that BALF CrAg can provide added diagnostic value in PC cases when the patient has a negative serum CrAg result but with strong suspicion of PC. There is emphasis placed on the potential utility of BALF CrAg when there is high suspicion for PCR and a negative serum CrAg result. To this end, if there were specific patients with BALF+/Serum- results, it would be recommended to expand upon these individual patient cases. Diagnostic methodologies continue to be evaluated given the challenges in obtaining a definitive diagnosis for pulmonary cryptococcosis.

Major Concerns:

1. There is emphasis placed on the potential utility of BALF CrAg testing when there is high suspicion for PC, yet a negative serum CrAg result. Please include specific details about CrAg BALF+/Serum- patient cases that may illustrate the utility of this testing.
2. As it relates to bronchoscopy fluid sampling from the patients in this study, please include further information about whether BALF sampling occurred consistent with the anatomical location of the suspicious radiological imaging and/or locations of nodules/lesions (i.e., left vs. right lung sidedness and/or combined BALF sampling for patients with bilateral lung involvement, etc.)
3. Please include a paragraph describing the limitations of this study.
4. It is recommended to review and to improve the cohesiveness of the overall manuscript to improve ease at which the reader can understand and interpret the data that is being presented.

Minor Concerns:

1. Lines 94-96: include references supporting this sentence.
2. Line 211: "...history of avian exposure" meaning what and in what context? This needs to be described further in the manuscript. If it is exposure to pigeons, please further describe in what context.
3. Lines 256-257: Further describe why CT results indicating early-stage disease and positive BALF CrAg tests was surprising for the 2 patients that were referenced (presumably this is because the parallel serum CrAg titer was considered to be 'weak'). Please specify in the text.
4. Lines 313-323: It is described that lumbar puncture was performed on any patient with neurological symptoms and/or positive serum CrAg-LFA. It is described that only 34 PC patients underwent an LP puncture due to various limitations. How many of the 42 PC patients did not meet the clinical/diagnostic criteria for LP to have been performed? Or did all 42 PC patients meet the described criteria?
5. Lines 322-323: How does the data described in this paragraph highlight the ability to detect cryptococcosis earlier with the advancement of CrAg-LFA testing technology?
6. Table 1: it is recommended to specify what is being displayed:
 - Age: mean/median and standard deviation?
 - Sex, male: female: may want to specify something like the following, 'n=19:23'
 - Leukocytosis: list units
 - Laboratory findings: presumably for leukocytosis and elevated CRP, this is taken out of the total 42 CP patients (meaning that all 42 of these patients had those labs taken?); would suggest that the denominator be included, since it is included for everything else under the 'Laboratory findings' heading.
 - CT images: Is 'Diffuse inflammatory shadows' and 'Diffuse inflammatory shadows with nodules' mutually exclusive?

Avian exposure history: please further define what is meant by this in the legend (response to reviewer comments define this as pigeon exposure-please further expand what is meant by this).

7. Table 2: It is suggested to adjust "Confirmed (n)" to something like "Confirmed PC Diagnosis (n)".

8. Table 3:

Legend: change "wake-positive" to "weak-positive"

Please confirm if this table is listing all PC patient results which had generated a weak serum CrAg and/or a weak BALF CrAg or is this just displaying the weak positive results which had a consistent pattern between serum CrAg result and BALF CrAg result.

Please describe why patients #8 and #9 were described as being false positives given that in the Table 3 title it is listed that these results are from the "42 PC patients". I thought the 42 PC patients were the ones confirmed to have a PC diagnosis based on the described criteria.

Reviewer #3 (Comments for the Author):

The revised version of the manuscript shows significant improvement compared with the previous draft. The authors have carefully addressed many of the earlier concerns, expanded methodological descriptions, clarified key points in the results and discussion, and refined the structure of the paper. As a result, the overall quality and readability of the manuscript have been markedly enhanced. However, several minor issues still need attention.

1. Articles ("the", "a", "an") are sometimes missing. Such as, "Diagnosis of pulmonary cryptococcosis remains challenge" should be "remains a challenge".
2. Use "sensitivity and specificity were" instead of "sensitivity and specificity was."
3. P-values should follow journal style: either $p = 0.05$ or $P = 0.05$, but keep uniform.

The revised version of the manuscript shows significant improvement compared with the previous draft. The authors have carefully addressed many of the earlier concerns, expanded methodological descriptions, clarified key points in the results and discussion, and refined the structure of the paper. As a result, the overall quality and readability of the manuscript have been markedly enhanced. However, several minor issues still need attention before final acceptance.

1. Articles ("the", "a", "an") are sometimes missing. Such as, "Diagnosis of pulmonary cryptococcosis remains challenge" should be "remains a challenge".
2. Use "sensitivity and specificity were" instead of "sensitivity and specificity was."
3. P-values should follow journal style: either $p = 0.05$ or $P = 0.05$, but keep uniform.

Reviewer #1 (Comments for the Author):

The authors present a study of the performance of a lateral flow cryptococcal antigen test using BAL fluid in non-HIV patients. The authors have either incorporated or addressed the prior reviewers' comments. I have only minor suggestions (line numbers refer to the Marked Up version):

Line 92: Incomplete sentence. Please rephrase for grammar.

Reply

Thank you. We have addressed and corrected the issues you highlighted in the revised manuscript. Additionally, we have enlisted the expertise of native English-speaking professionals to further refine the entire text, ensuring it meets the standards for publication.

Line 304: Remove the word "podocarp" as it is not widely used or understood.

Reply

Thank you. We have removed the word "podocarp".

Lines 231-233: The Table 2 definition of a combined positive is clear. Please add the same description to the manuscript.

Reply

Thank you for your suggestions. We have added the definition of a combined positive on line 231-232 in the revised manuscript.

Table 2: Please explain what "confirmed" means either in the legend of Table 2 or in the manuscript.

Reply

Thank you. The word "confirmed" in this context is expressed as "The final clinical diagnosis was confirmed as pulmonary cryptococcosis."

Table 3: Correct spelling of "weak" in the legend.

Reply

Corrected.

Reviewer #2 (Comments for the Author):

Summary of Key Findings:

This study sought to assess the performance and diagnostic value of the cryptococcal antigen lateral flow immunochromatographic assay (CrAg-LFA) (IMMY, USA) on bronchoalveolar lavage fluid (BALF) from patients with pulmonary cryptococcosis (PC) in comparison to serum. This prospective analysis included patients suspected of having pulmonary cryptococcosis (PC) receiving treatment at a tertiary Beijing hospital from Jan. 2013-Feb. 2024. 514 non-HIV patients with characteristic PC imaging and no prior antifungal therapy were included; 42 of these patients were ultimately diagnosed with PC through various means with criteria described herein. BALF CrAg-LFA had 88.1% sensitivity, 99.6% specificity, 94.9% PPV, and 98.9% NPV-values similar to the parallel serum CrAg-LFA. The authors state that combining BALF CrAg-LFA and serum CrAg-LFA results (positive result define as BALF+ and/or Serum+) increases sensitivity and NPV to 100% and suggest that BALF CrAg can provide added diagnostic value in PC cases when the patient has a negative serum CrAg result but with strong suspicion of PC. There is emphasis placed on the potential utility of BALF CrAg when there is high suspicion for PCR and a negative serum CrAg result. To this end, if there were specific patients with BALF+/Serum- results, it would be recommended to expand upon these individual patient cases. Diagnostic methodologies continue to be evaluated given the challenges in obtaining a definitive diagnosis for pulmonary cryptococcosis.

Reply

Thank you for your positive comments.

Major Concerns:

1. There is emphasis placed on the potential utility of BALF CrAg testing when there is high suspicion for PC, yet a negative serum CrAg result. Please include specific details about CrAg BALF+/Serum- patient cases that may illustrate the utility of this testing.

Reply

We sincerely thank the reviewer for this insightful comment. In our study, we identified four cases with discordant CrAg results, specifically negative serum CrAg but positive BALF-CrAg findings. Among these cases, three patients were asymptomatic at the time of clinical evaluation, with chest CT imaging predominantly revealing solitary pulmonary nodules. The fourth case involved an immunocompromised patient who presented with a 15-day history of persistent fever. This patient exhibited a weakly positive BALF-CrAg result,

and chest CT imaging demonstrated localized patchy consolidation in the right lower lobe. These cases underscore the clinical utility of BALF-CrAg testing in scenarios where there is a high suspicion of pulmonary cryptococcosis (PC) despite negative serum CrAg results.

We have incorporated this detailed description in the revised manuscript (Lines 312-316) to provide a more comprehensive analysis of these findings.

2. As it relates to bronchoscopy fluid sampling from the patients in this study, please include further information about whether BALF sampling occurred consistent with the anatomical location of the suspicious radiological imaging and/or locations of nodules/lesions (i.e., left vs. right lung sidedness and/or combined BALF sampling for patients with bilateral lung involvement, etc.)

Reply

Thank you. BALF was collected from patients after chest CT-guided localization. Therefore, we believe that the site of BALF collection corresponds to the lesion identified by CT. We have added this description on line 133-138 in the revised manuscript.

3. Please include a paragraph describing the limitations of this study.

Reply

We sincerely thank the reviewer for their valuable feedback. We acknowledge the limitations of our study, which include the relatively small sample size of confirmed pulmonary cryptococcosis (PC) cases (n = 42) and the single-center design. These factors may introduce selection bias and limit the generalizability of our findings. To address these limitations, we strongly recommend that future studies adopt a multi-center design and enroll a larger cohort of PC cases to validate and extend our results. We have incorporated this acknowledgment and recommendation in the conclusions section of the revised manuscript to provide a more balanced and comprehensive discussion of our study's implications.

4. It is recommended to review and to improve the cohesiveness of the overall manuscript to improve ease at which the reader can understand and interpret the data that is being presented.

Reply

Thank you for your suggestions. We have revised the manuscript to make the findings clearer and easier to understand.

Minor Concerns:

1. Lines 94-96: include references supporting this sentence.

Reply

Thank you. We have added the reference (PMID: 30329097) on line 94 in the revised manuscript.

2. Line 211: "...history of avian exposure" meaning what and in what context? This needs to be described further in the manuscript. If it is exposure to pigeons, please further describe in what context.

Reply

Thank you. "A history of avian exposure" refers to a patient's exposure to pigeons, either through keeping them at home or in the neighborhood, or by regularly visiting places where pigeons gather.

We have included a description of this point at line 209-210 in the revised manuscript.

3. Lines 256-257: Further describe why CT results indicating early-stage disease and positive BALF CrAg tests was surprising for the 2 patients that were referenced (presumably this is because the parallel serum CrAg titer was considered to be 'weak'). Please specify in the text.

Reply

Thank you. Out of the 12 patients with weakly positive serum CrAg results, 10 showed weakly positive or negative BALF-CrAg results; the remaining two patients, despite exhibiting positive BALF CrAg tests, were asymptomatic, with chest CT scans showing solitary nodules indicative of early-stage disease. We have modified the description on line 258-261 in the revised manuscript.

4. Lines 313-323: It is described that lumbar puncture was performed on any patient with neurological symptoms and/or positive serum CrAg-LFA. It is described that only 34 PC patients underwent an LP puncture due to various limitations. How many of the 42 PC patients did not meet the clinical/diagnostic criteria for LP to have been performed? Or did all 42 PC patients meet the described criteria?

Reply

Thank you. In this study, we performed lumbar puncture with CSF analysis in PC patients who met any of the following criteria: 1) Neurological symptoms (fever, headache,

nausea/vomiting, or altered mental status); 2) Positive serum CrAg-LFA. Ultimately, 39/42 PC patients met the criteria. Due to various limitations (such as technical challenges, bleeding disorders, and patient refusal), CSF analysis (including culture, India ink staining, and CrAg-LFA testing) was performed in 34 PC patients.

We have added the description on line 325 in the revised manuscript.

5. Lines 322-323: How does the data described in this paragraph highlight the ability to detect cryptococcosis earlier with the advancement of CrAg-LFA testing technology?

Reply

Thank you. In this study, the low detection rate of *Cryptococcus* in the CSF suggests that most patients were at an early stage of infection. This finding further emphasizes that the CrAg-LFA test, owing to its high sensitivity and specificity, offers a valuable advantage over conventional methods in facilitating the early diagnosis of cryptococcosis.

We have modified the description on line 330-333 in the revised manuscript.

6. Table 1: it is recommended to specify what is being displayed:

- Age: mean/median and standard deviation?
- Sex, male: female: may want to specify something like the following, 'n=19:23'
- Leukocytosis: list units
- Laboratory findings: presumably for leukocytosis and elevated CRP, this is taken out of the total 42 CP patients (meaning that all 42 of these patients had those labs taken?); would suggest that the denominator be included, since it is included for everything else under the 'Laboratory findings' heading.
- CT images: Is 'Diffuse inflammatory shadows' and 'Diffuse inflammatory shadows with nodules' mutually exclusive?
- Avian exposure history: please further define what is meant by this in the legend (response to reviewer comments define this as pigeon exposure-please further expand what is meant by this).

Reply

Thank you for your suggestions. In this study, the age data followed a normal distribution and were therefore expressed as mean and standard deviation ($\bar{x} \pm s$). Furthermore, the CT feature classification "diffuse inflammatory shadows" indicates the presence of shadows alone, while "diffuse inflammatory shadows with nodules" indicates both findings. These two categories are mutually exclusive.

We have revised the result section and table 1 to incorporate the suggested revisions in the revised manuscript.

7. Table 2: It is suggested to adjust "Confirmed (n)" to something like "Confirmed PC Diagnosis (n)".

Reply

Thank you for your suggestions. We have updated a description of this point in Table 2 of the revised manuscript.

8. Table 3:

- Legend: change "wake-positive" to "weak-positive"
- Please confirm if this table is listing all PC patient results which had generated a weak serum CrAg and/or a weak BALF CrAg or is this just displaying the weak positive results which had a consistent pattern between serum CrAg result and BALF CrAg result.
- Please describe why patients #8 and #9 were described as being false positives given that in the Table 3 title it is listed that these results are from the "42 PC patients". I thought the 42 PC patients were the ones confirmed to have a PC diagnosis based on the described criteria.

Reply

Thank you for your suggestions. Table 3 specifically displays the weak CrAg intensity results demonstrating a consistent pattern between serum CrAg results and BALF CrAg results, and it includes false-positive cases.

We have revised Table 3 accordingly in our updated manuscript to address all the points raised.

Reviewer #3 (Comments for the Author):

The revised version of the manuscript shows significant improvement compared with the previous draft. The authors have carefully addressed many of the earlier concerns, expanded methodological descriptions, clarified key points in the results and discussion, and refined the structure of the paper. As a result, the overall quality and readability of the manuscript have been markedly enhanced. However, several minor issues still need attention.

1. Articles ("the", "a", "an") are sometimes missing. Such as, "Diagnosis of pulmonary cryptococcosis remains challenge" should be "remains a challenge".

2. Use "sensitivity and specificity were" instead of "sensitivity and specificity was."
3. P-values should follow journal style: either $p = 0.05$ or $P = 0.05$, but keep uniform.

Reply

Thank you. We have revised the manuscript to address all the points raised.

Re: Spectrum01083-25R1 (Evaluation of a lateral flow immunochromatographic assay for detecting cryptococcal antigens in bronchoalveolar lavage fluid from HIV-negative patients with pulmonary cryptococcosis)

Dear Prof. Hui Wang:

Thank you for the privilege of reviewing your work. Below you will find my comments, instructions from the Spectrum editorial office, and the reviewer comments.

Revision Guidelines

Sincerely,
Dhammika Navarathna
Editor
Microbiology Spectrum

Reviewer #2 (Comments for the Author):

Summary of Key Findings:

The authors have made significant improvements to the manuscript in response to this round of reviewers' comments. There are minor concerns suggested that should be addressed.

Minor Concerns (line numbers corresponding to Spectrum01083-25R1-Marked_Up_Manuscript pdf file):

1. Grammar/Editing

Line 33: Either change back to "...over 11 years in Beijing, China..." or change to, "...over an 11-year period in Beijing, China..."

Line 44: Suggest adjustment to, "...in this cohort of HIV-negative PC patients."

Lines 200-202: suggested changes below, including breaking up into two sentences.

Sentence 1: Twenty-five (59.5%) patients were symptomatic.

Sentence 2: The most common symptoms included fever, cough, dyspnea, and headache; altered mental status and hematuria were occasionally observed.

*Please also confirm if "hematuria" is meant or "hemoptysis" is meant.

Line 222: remove "an" from the phrase, "...had no an avian exposure history, whereas most patients with avian exposure..."

The manuscript references "hematuria" three times throughout the manuscript/legends as a clinical sign. Is this truly hematuria (blood in the urine), or is this meant to be hemoptysis (blood being coughed up)? Please make edits as necessary.

Lines 287-288: adjust to, "...detects free cryptococcal capsular antigen in body fluids such as blood and CSF."

Line 299: change "researches" back to "research"

Lines 310-314: Change to two separate sentences:

Sentence 1: One possible explanation is that diffuse inflammatory shadows were observed in 33 (78.6%) of the PC patients in our cohort.

Sentence 2: CrAg derived from such diffuse lesions may be more likely to enter the bloodstream compared to localized nodular lesions, facilitating easier detection in serum.

2. Line 58: States that "...(combination) BALF And serum tests increased specificity and positive predictive value to 100%..."

This contrasts with the wording presented in the following sections of the paper: the abstract, lines 245-246, lines 319-320, lines 376-377.

3. Line 258: Is weakly positive BALF CrAg samples supposed to be "5/5, 100%" or "4/5, 80%"? Like what is described in lines 256-257?

4. Lines 360-362: This sentence does not make sense within the paragraph and deletion of the following sentence is recommended: "This finding further emphasizes that the CrAg-LFA, owing to its high sensitivity and specificity, over conventional methods in facilitating the early diagnosis of cryptococcosis."

5. Lines 380-381: would adjust to the following: "...we recommend adding a BALF CrAg test to facilitate the diagnosis of PC..."

Lines 377-382: would suggest breaking up this section into a couple sentences.

6. Line 531: The legend should be adjusted to accurately reflect what the figure is depicting: "Conversely, among symptomatic patients, 52% had serum CrAg levels \leq 1:40."

Reviewer #3 (Comments for the Author):

The authors have satisfactorily addressed all previous concerns. The manuscript has been significantly improved and is now suitable for publication.

Reviewer #2 (Comments for the Author):

Summary of Key Findings:

The authors have made significant improvements to the manuscript in response to this round of reviewers' comments. There are minor concerns suggested that should be addressed.

Minor Concerns (line numbers corresponding to Spectrum01083-25R1-Marked_Up_Manuscript pdf file):

1. Grammar/Editing

- Line 33: Either change back to "...over 11 years in Beijing, China..." or change to, "...over an 11-year period in Beijing, China..."
 - Line 44: Suggest adjustment to, "...in this cohort of HIV-negative PC patients."
 - Lines 200-202: suggested changes below, including breaking up into two sentences.
 - Sentence 1: Twenty-five (59.5%) patients were symptomatic.
 - Sentence 2: The most common symptoms included fever, cough, dyspnea, and headache; altered mental status and hematuria were occasionally observed.
- Line 222: remove "an" from the phrase, "...had no an avian exposure history, whereas most patients with avian exposure..."

Reply

Thank you. We have corrected as you suggested. Please line 33, line 43, line 198-200, line 212 in the clean version.

*Please also confirm if "hematuria" is meant or "hemoptysis" is meant. The manuscript references "hematuria" three times throughout the manuscript/legends as a clinical sign. Is this truly hematuria (blood in the urine), or is this meant to be hemoptysis (blood being coughed up)? Please make edits as necessary.

Reply

This patient has a 20-year history of intermittent fever and systemic erythema and sought medical attention for 2 months due to fever and hematuria. Laboratory testing of the patient's urine using the cryptococcal antigen lateral flow assay (CrAg-LFA) yielded a positive result. Subsequently, serum and bronchoalveolar lavage fluid (BALF) were also tested for CrAg-LFA, leading to a definitive diagnosis of pulmonary cryptococcosis (PC).

- Lines 287-288: adjust to, "...detects free cryptococcal capsular antigen in body fluids such as blood and CSF."
- Line 299: change "researches" back to "research"
- Lines 310-314: Change to two separate sentences:
- Sentence 1: One possible explanation is that diffuse inflammatory shadows were observed in 33 (78.6%) of the PC patients in our cohort.
- Sentence 2: CrAg derived from such diffuse lesions may be more likely to enter the bloodstream compared to localized nodular lesions, facilitating easier detection in serum.

Reply

Thank you. We have corrected as you suggested. Please line 273-274, line 281, line 291-294 in the clean version.

2. Line 58: States that "...(combination) BALF And serum tests increased specificity and positive predictive value to 100%...". This contrasts with the wording presented in the following sections of the paper: the abstract, lines 245-246, lines 319-320, lines 376-377.

Reply

Thank you. We have updated line 58 to "BALF and serum tests increased sensitivity and negative predictive value to 100%" to align with the parallel phrasing used elsewhere in the manuscript, as you rightly pointed out.

3. Line 258: Is weakly positive BALF CrAg samples supposed to be "5/5, 100%" or "4/5, 80%"? Like what is described in lines 256-257?

Reply

Thank you. We intended to convey that 80% false-negative (4/5) and 100% weakly positive (5/5) BALF CrAg results were associated with serum CrAg titers $\leq 1:20$. As suggested, we have revised the manuscript to make the findings clearer and easier to understand.

4. Lines 360-362: This sentence does not make sense within the paragraph and deletion of the following sentence is recommended: "This finding further emphasizes that the CrAg-LFA, owing to its high sensitivity and specificity, over conventional methods in facilitating the early diagnosis of cryptococcosis."

Reply

Thank you for the suggestion. We have removed the indicated sentence as recommended.

5. Lines 380-381: would adjust to the following: "...we recommend adding a BALF CrAg test to facilitate the diagnosis of PC..."

Lines 377-382: would suggest breaking up this section into a couple sentences.

Reply

Thank you for your input. We have updated the section to: "We recommend adding a BALF CrAg test in certain clinical scenarios, based on the premise that BALF accurately reflects lung infection status (especially in cases with small or localized lesions). This is particularly useful for facilitating the diagnosis of PC when serological tests are negative, such as when patients are already undergoing bronchoscopy for therapy or other diagnostics."

6. Line 531: The legend should be adjusted to accurately reflect what the figure is depicting: "Conversely, among symptomatic patients, 52% had serum CrAg levels $\leq 1:40$."

Reply

Thank you. As requested, we have revised the figure legend to: "The figure demonstrates the distribution of serum CrAg titers in asymptomatic and symptomatic patients. Asymptomatic patients were significantly more likely to exhibit low serum CrAg titers ($\leq 1:40$) than

symptomatic patients (82% vs. 52%; $p=0.044$)."

Reviewer #3 (Comments for the Author):

The authors have satisfactorily addressed all previous concerns. The manuscript has been significantly improved and is now suitable for publication.

Reply

Thank you.

Re: Spectrum01083-25R2 (Evaluation of a lateral flow immunochromatographic assay for detecting cryptococcal antigens in bronchoalveolar lavage fluid from HIV-negative patients with pulmonary cryptococcosis)

Dear Prof. Hui Wang:

Your manuscript has been accepted, and I am forwarding it to the ASM production staff for publication. Your paper will first be checked to make sure all elements meet the technical requirements. ASM staff will contact you if anything needs to be revised before copyediting and production can begin. Otherwise, you will be notified when your proofs are ready to be viewed.

Sincerely,
Dharmika Navarathna
Editor
Microbiology Spectrum

Reviewer #2 (Comments for the Author):

The authors have addressed the reviewers' comments adequately.